# GENERALIZATION AND REGULARIZATION IN DQN

## ABSTRACT

Deep reinforcement learning (RL) algorithms have shown an impressive ability to learn complex control policies in high-dimensional environments. However, despite the ever-increasing performance on popular benchmarks like the Arcade Learning Environment (ALE), policies learned by deep RL algorithms can struggle to generalize when evaluated in remarkably similar environments. These results are unexpected given the fact that, in supervised learning, deep neural networks often learn robust features that generalize across tasks. In this paper, we study the generalization capabilities of DQN in order to aid in understanding this mismatch between generalization in deep RL and supervised learning methods. We provide evidence suggesting that DQN overspecializes to the domain it is trained on. We then comprehensively evaluate the impact of traditional methods of regularization from supervised learning, $\ell_2$ and dropout, and of reusing learned representations to improve the generalization capabilities of DQN. We perform this study using different game modes of Atari 2600 games, a recently introduced modification for the ALE which supports slight variations of the Atari 2600 games used for benchmarking in the field. Despite regularization being largely underutilized in deep RL, we show that it can, in fact, help DQN learn more general features. These features can then be reused and fine-tuned on similar tasks, considerably improving the sample efficiency of DQN.

## 1 INTRODUCTION

Recently, reinforcement learning (RL) has proven very successful on complex high-dimensional problems, in large part due to the increase in computational power and to the use of deep neural networks for function approximation (e.g., Mnih et al., 2015; Silver et al., 2016). Despite the generality of the proposed solutions, applying these algorithms to slightly different environments generally requires agents to learn the task from scratch. Practitioners often realize that the learned policies rarely generalize to other domains, even when they are remarkably similar, and that the learned representations are seldom reusable.

Deep neural networks, though, are lauded for their generalization capabilities (e.g., LeCun et al., 1998). Some communities heavily rely on reusing representations learned by neural networks. In computer vision, classification and segmentation algorithms are rarely trained from scratch; instead they are initialized with pre-trained models from larger datasets like ImageNet (e.g., Razavian et al., 2014; Long et al., 2015). The field of natural language processing has also seen successes in reusing and refining weights from certain layers of neural networks using pre-trained word embeddings, with more recent techniques able to reuse all weights of the network (e.g., Howard & Ruder, 2018).

In light of the successes of traditional supervised learning methods, the current lack of generalization or reusable knowledge (e.g., policies, representation) acquired by current deep RL algorithms is somewhat surprising. In this paper we investigate whether the representation learned by deep RL methods can be generalized, or at the very least reused and refined on small variations to the task at hand. First, we evaluate the generalization capabilities of DQN (Mnih et al., 2015). We further explore whether the experience gained by the supervised learning community to improve generalization and to avoid overfitting could be used in deep RL. We employ conventional supervised learning techniques, albeit largely unexplored in deep RL, such as fine-tuning (i.e., reusing and refining the representation) and regularization. We show that a learned representation trained with regularization allows us to learn more general features capable of being reused and fine-tuned. Besides improving the generalization capabilities of the learned policies this fine-tuning procedure has the potential to

greatly improve sample efficiency on settings in which an agent might face multiple variations of the same task. Finally, the results we present here also can be seen as paving a way towards novel curriculum learning approaches for deep RL.

We perform our experiments using different game modes and difficulties of Atari 2600 games, a newly introduced feature of the Arcade Learning Environment (ALE; Bellemare et al., 2013). These game modes allow agents to be trained in one environment while being evaluated in a slightly different environment that still captures key concepts of the original environment (e.g., game sprites, agent goals, dynamics). This use of game modes is itself a novel approach for measuring our progress toward a longstanding goal of agents that can learn to be generally competent and generalize across tasks (Bellemare et al., 2013; Machado et al., 2018; Nichol et al., 2018). This paper also introduces the first baselines for the different modes of Atari 2600 games.

## 2 BACKGROUND

### 2.1 REINFORCEMENT LEARNING

Reinforcement learning (RL) is a problem where an agent interacts with an environment with the goal of maximizing some form of cumulative long term reward. RL problems are often modeled as a Markov decision process (MDP), defined by a 5-tuple $\langle \mathcal{S}, \mathcal{A}, p, r, \gamma \rangle$. At a discrete time step $t$ the agent observes the current state $S_t \in \mathcal{S}$ and chooses an action $A_t \in \mathcal{A}$ to probabilistically transition to the next state $S_{t+1} \in \mathcal{S}$ according to the transition dynamics function $p(s' \,|\, s, a) \doteq P(S_{t+1} = s' \,|\, S_t = s\,, A_t = a)$. The agent receives a reward signal $R_{t+1}$ according to the reward function $r : \mathcal{S} \times \mathcal{A} \to \mathbb{R}$. The agents goal is to learn a policy $\pi : \mathcal{S} \times \mathcal{A}$ defined as the conditional probability of taking action $a$ in state $s$ written as $\pi(a \,|\, s)$. The learning agent refines its policy with the objective of maximizing the expected return, that is, the cumulative discounted reward incurred from time $t$, defined by $G_t \doteq \sum_{k=0}^{\infty} \gamma^k R_{t+k+1}$ where $\gamma \in [0, 1)$ is the discount factor.

Q-learning (Watkins & Dayan, 1992) is a traditional approach to learning an optimal policy from samples obtained from interactions with the environment. It is used to learn an optimal state-action value function via a bootstrapped iterative method. For a given policy $\pi$ we define the state-action value function as the expected return conditioned on a state and action $q_\pi(s, a) \doteq \mathbb{E}_\pi \big[ G_t | S_t = s, A_t = a \big]$. The agent iteratively updates the state-action value function based on samples from the environment using the update rule

$$Q(S_t, A_t) \leftarrow Q(S_t, A_t) + \alpha \big[ R_{t+1} + \gamma \max_{a' \in \mathcal{A}} Q(S_{t+1}, a') - Q(S_t, A_t) \big]$$

where $t$ denotes the current timestep and $\alpha$ the step size. Generally, due to the exploding size of the state space in many real-world problems, it is intractable to learn a state-action pairing for the entire MDP, with researchers and practitioners often resorting to learning an approximate to $q_\pi$.

DQN approximates the state-action value function such that $q_\pi(s, a) \approx Q(s, a; \theta)$, where $\theta$ denotes the weights of a neural network. The network takes as input some encoding of the current state $S_t$ and outputs $|\mathcal{A}|$ scalars corresponding to the state-action values for that given state. DQN is trained to minimize

$$L^{\text{DQN}} = \mathbb{E}_{S_t, A_t, R_{t+1}, S_{t+1} \sim U(\cdot)} \big[ \big( R_{t+1} + \max_{a' \in \mathcal{A}} Q(S_{t+1}, a'; \theta^-) - Q(S_t, A_t; \theta) \big)^2 \big]$$

where $(S_t, A_t, R_{t+1}, S_{t+1})$ are uniformly sampled from $U(\cdot)$, the experience replay buffer filled with experience collected by the agent. The weights $\theta^-$ of a duplicate network are updated less frequently for stability purposes.

### 2.2 SUPERVISED LEARNING

In the supervised learning problem we are given a dataset of examples represented by a matrix $X \in \mathbb{R}^{m \times n}$ with $m$ training examples of dimension $n$, and a vector $\mathbf{y} \in \mathbb{R}^{1 \times m}$ denoting the output target $y_i$ for each training example $X_i$. We want to learn a function which maps each training example $X_i$ to its predicted output label $\hat{y}_i$. The goal is to learn a robust model that accurately predicts $y_i$ from $X_i$ while also being able to generalize to unseen training examples. In this paper

we focus on using a neural network parameterized by the weights $\theta$ to learn the function $f$ such that $\hat{y}_i = f(X_i; \theta)$. We typically train these models by minimizing

$$\min_{\theta} \; \frac{\lambda}{2} \, \|\theta\|_2^2 + \frac{1}{m} \sum_{i=1}^{m} L(y_i, \hat{y}_i) = \min_{\theta} \; \frac{\lambda}{2} \, \|\theta\|_2^2 + \frac{1}{m} \sum_{i=1}^{m} L(y_i, f(X_i; \theta))$$

where $L$ is a differentiable loss function which outputs a scalar determining the quality of the prediction (e.g., squared error loss). The first term is a form of regularization, i.e., $\ell_2$ regularization, which encourages generalization. $\ell_2$ regularization imposes a penalty on large weight vectors with $\lambda$ being the weighted importance of the regularization term.

Another popular regularization technique is dropout (Srivastava et al., 2014). When using dropout, during forward propagation each neural unit has a chance of being set to zero according to a Bernoulli distribution with probability $p \in [0, 1]$, referred to as the dropout rate. Dropout discourages the network from relying on a small number of neurons to make a prediction, making it hard for the network to memorize the dataset.

Prior to training, the network parameters are usually initialized through a stochastic process (e.g., Xavier initialization; Glorot & Bengio, 2010). We can also initialize the network using pre-trained weights from a different task. If we reuse one or more pre-trained layers we say the weights encoded by those layers will be fine-tuned during training (e.g., Razavian et al., 2014; Long et al., 2015).

## 3   THE ALE AS A PLATFORM FOR EVALUATING GENERALIZATION

The Arcade Learning Environment (ALE) is a platform used to evaluate agents across dozens of Atari 2600 games (Bellemare et al., 2013). It has become one of the standard evaluation platforms in the field and has led to a number of exciting algorithmic advances (e.g., Mnih et al., 2015). The ALE poses the problem of general competency by having agents use the same learning algorithm to perform well in as many games as possible, while learning without using game specific knowledge. Learning to play multiple games with the same agent, or learning to play a game faster by leveraging knowledge acquired in a different game is much harder, with fewer successes being known (e.g., Rusu et al., 2016; Kirkpatrick et al., 2016; Parisotto et al., 2016; Schwarz et al., 2018; Espeholt et al., 2018).

In this paper, we use the different modes and difficulties of Atari 2600 games to evaluate a neural network's ability to generalize in high-dimensional state spaces. Game modes, originally native to the Atari console, were recently added in the ALE (Machado et al., 2018). They give us modifications of the default environment dynamics and state space, often modifying sprites, velocities, and partial observability. These modes pose a tractable way to investigate generalization of RL agents in a high-dimensional environment. Instead of requiring an agent to play multiple games that are visually very different or even non-analogous, it requires agents to play games that are visually very similar and that can be played with policies that are very similar, at least from a human perspective.

We use 13 flavours (combinations of a mode and a difficulty) obtained from 4 games: FREEWAY, HERO, BREAKOUT, and SPACE INVADERS. In FREEWAY, the different modes vary the speed and number of vehicles, while different difficulties change how the player is penalized for running into a vehicle. In HERO, subsequent modes start the player off at increasingly harder levels of the game. The mode we use in BREAKOUT makes the bricks partially observable. The used modes in SPACE INVADERS allow for oscillating shield barriers, increasing the width of the player sprite, and partially observable aliens. Full explanations of specific games, their modes, and their difficulties can be found in Appendix A. Figure 1 provides screenshots showing side by side comparisons of some of the modes explored in this paper. When reading the analyses of this paper it is important to keep in mind how remarkably similar these modes are.

## 4   GENERALIZATION OF THE LEARNED POLICIES AND OVERFITTING

In order to test the generalization capabilities of DQN we first evaluate whether a policy learned in one flavour can perform well in a different flavour. As afformentioned, different modes and difficulties of a single game look very similar. If the representation encodes a robust policy we might expect it to be able to generalize to slight variations of the underlying reward signal, game

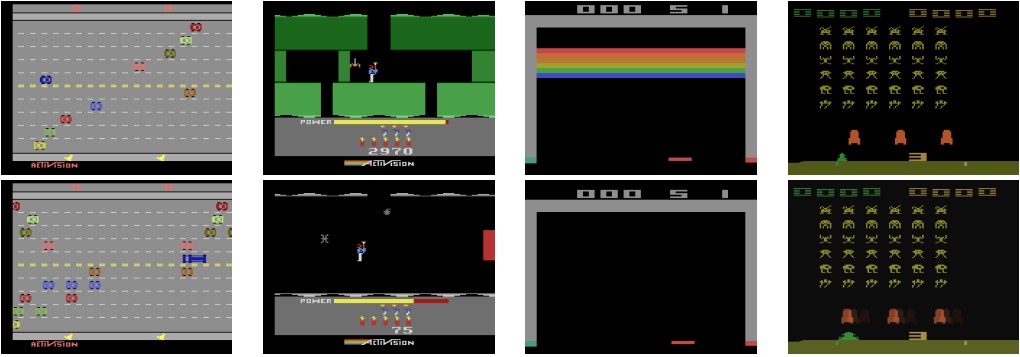

Figure 1: Each column shows variation between two selected flavours of each game. From left to right: FREEWAY, HERO, BREAKOUT, and SPACE INVADERS.

dynamics, or observations. Evaluating the learned policy in a similar but different flavour can be seen as evaluating generalization in RL, similar to cross-validation in supervised learning.

To evaluate DQN's ability to generalize across flavours we evaluate the learned $\epsilon$-greedy policy on a new flavour after being trained for 50M frames in the default flavour, m0d0 (mode 0, difficulty 0). We measure the cumulative reward averaged over 100 episodes in the new flavour, adhering to the evaluation protocol suggested by Machado et al. (2018). The results are summarized in Table 1. Baseline results where the agent is trained from scratch for 50M frames in the flavour we use for evaluation are summarized in the baseline column. Theoretically, this baseline can be seen as an upper bound on the performance DQN can achieve in that flavour, as it represents the agent's performance when evaluated in the same flavour it was trained on. Full baseline results with the agent's performance after different number of frames can be found in Appendix B.

We can see in the results that the policies learned by DQN do not generalize well to different flavours, even when the flavours are remarkably similar. For example, in FREEWAY, a high-level policy applicable to all flavours is to go up while avoiding cars. Perhaps surprisingly, this does not seem to be what DQN learns. For example, the default flavour m0d0 and m4d0 have exactly the same sprites on the screen, the only difference is that in m4d0 some cars accelerate and decelerate over time. The close to optimal policy learned in m0d0 is only able to score 15.8 points when evaluated on m4d0, which is approximately half of what the policy learned from scratch in that flavour achieves (29.9 points). The learned policy when evaluated on flavours that differ more from m0d0 perform even worse.

As previously mentioned, the different modes of HERO can be seen as giving the agent a curriculum or a natural progression. Interestingly, the agent trained in the default mode for 50M frames can progress to at least level 3 and sometimes level 4. Mode 1 starts the agent off at level 5, and performance in this mode suffers greatly during evaluation. There are very few game mechanics added to level 5, indicating that perhaps the agent is memorizing trajectories instead of learning a robust policy capable of solving each level.

The results in some flavours suggest that the agent is overfitting to the flavour it is trained on. We tested this hypothesis by periodically evaluating the policy being learned in each of the other flavours of that game. This process involved taking checkpoints of the network at every $500,000$ frames and evaluating the $\epsilon$-greedy policy in the prescribed flavour for 100 episodes, again further averaged over five runs. The obtained results in FREEWAY, the most pronounced game in which we see this overfitting trend, are depicted in Figure 2. Learning curves for all flavours can be found in Appendix C.

In FREEWAY, while we see the policy's performance flattening out in m4d0, we do see the traditional bell-shaped curve associated to overfitting in the other modes. At first, improvements in the original policy do correspond to improvements in the performance of that policy in other domains. With time, it seems that it starts to refine its policy for the specific flavour it is being trained on, overfitting to that flavour. With other game flavours being significantly more complex in their dynamics and gameplay, we do not observe this prominent bell-shaped curve though. For example, in BREAKOUT, we actually observe a monotonic increase in performance throughout the evaluation process.

| GAME VARIANT | | EVALUATION | | LEARN SCRATCH | |
|---|---|---|---|---|---|
| | m1d0 | 0.2 | (0.2) | 4.8 | (9.3) |
| FREEWAY | m1d1 | 0.1 | (0.1) | 0.0 | (0.0) |
| | m4d0 | 15.8 | (1.0) | 29.9 | (0.7) |
| HERO | m1d0 | 82.1 | (89.3) | 1425.2 | (1755.1) |
| | m2d0 | 33.9 | (38.7) | 326.1 | (130.4) |
| BREAKOUT | m12d0 | 43.4 | (11.1) | 67.6 | (32.4) |
| | m1d0 | 258.9 | (88.3) | 753.6 | (31.6) |
| SPACE INVADERS | m1d1 | 140.4 | (61.4) | 698.5 | (31.3) |
| | m9d0 | 179.0 | (75.1) | 518.0 | (16.7) |

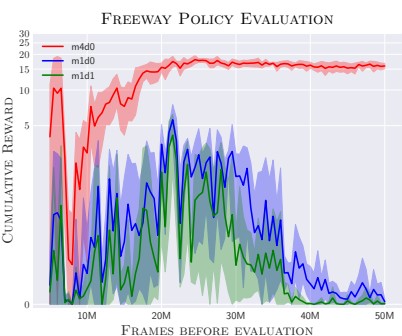

Table 1: Direct policy evaluation. Each game was initially trained in the default mode for 50M frames then evaluated in each listed game flavour. Reported numbers are the average over 5 runs. Standard deviation is reported between parentheses.

Figure 2: Performance of an agent that was trained in the default mode of FREEWAY and evaluated at every $500,000$ frames in each corresponding mode. Results are averaged over five seeds. The y-axis is log scaled.

In conclusion, when looking at Table 1, it seems that the policies learned by DQN struggle to generalize to even small variations encountered in game flavours. This lack of generalization is surprising, and results as seen in FREEWAY exhibit a troubling notion of overfitting. Based on these results we aim to evaluate whether deep RL could benefit from established methods from supervised learning promoting generalization and reducing overfitting.

# 5 REGULARIZATION IN DEEP RL

In order to evaluate the hypothesis that the observed lack of generalization is due to overfitting, we revisit some popular regularization methods from the supervised learning literature. The two forms of regularization we test are dropout and $\ell_2$ regularization.

First we want to understand the effect of regularization on evaluating the learned policy in a different flavour. We do so by applying dropout to the first four layers of the network during training, that is, the three convolutional layers and the first fully connected layer. We simultaneously apply $\ell_2$ regularization on all weights in the network based on preliminary experiments that showed an additive effect when combining dropout and $\ell_2$ regularization. This confirms, for example, Srivastava et al.'s (2014) result that these methods provide benefit in tandem.

We follow the same evaluation scheme described when evaluating the unregularized policy to different flavours. We evaluate the policy learned after 50M frames of the default mode of each game. A grid search was performed on FREEWAY to find reasonable hyperparameters for the dropout rate $p \in \{0.05, 0.1, 0.2, 0.3, 0.4, 0.5\}$ and the weighted regularization parameter $\lambda \in \{10^{-2}, 10^{-3}, 10^{-4}\}$. These parameters were then used for each subsequent flavour. Notably, significantly smaller dropout values were required compared to heuristics used in supervised learning, although this could be due to the small size of the network in question. We ended up choosing $\lambda = 10^{-4}$, $p = 0.05$ for the first three convolutional layers, and $p = 0.1$ for the first fully connected layer. We contrast these results with the results presented in the previous section. This evaluation protocol allows us to directly evaluate the effect of regularization on the learned policy's ability to generalize. A baseline agent trained from scratch for 50M frames in each flavour is also provided. The results are presented in Table 2 with the evaluation learning curves being available in the Appendix.

When using regularization during training we sometimes observe a performance hit in the default flavour. Dropout generally requires increased training iterations to reach the same level of performance sans-dropout. Suprisingly, we did not observe this performance hit in all games. Nevertheless, maximal performance in one flavour is not our goal. We are interested in the setting where one may be willing to take lower performance on one task in order to obtain higher performance, or adaptability, on future tasks. Nevertheless, full baseline results using regularization in the default flavour can also be found in Table 7 in the Appendix.

| GAME VARIANT | | EVAL. WITH REGULARIZATION | | EVAL. WITHOUT REGULARIZATION | | LEARN SCRATCH | |
|---|---|---|---|---|---|---|---|
| FREEWAY | m1d0 | 5.8 | (3.5) | 0.2 | (0.2) | 4.8 | (9.3) |
| | m1d1 | 4.4 | (2.3) | 0.1 | (0.1) | 0.0 | (0.0) |
| | m4d0 | 20.6 | (0.7) | 15.8 | (1.0) | 29.9 | (0.7) |
| HERO | m1d0 | 116.8 | (76.0) | 82.1 | (89.3) | 1425.2 | (1755.1) |
| | m2d0 | 30.0 | (36.7) | 33.9 | (38.7) | 326.1 | (130.4) |
| BREAKOUT | m12d0 | 31.0 | (8.6) | 43.4 | (11.1) | 67.6 | (32.4) |
| SPACE INVADERS | m1d0 | 456.0 | (221.4) | 258.9 | (88.3) | 753.6 | (31.6) |
| | m1d1 | 146.0 | (84.5) | 140.4 | (61.4) | 698.5 | (31.3) |
| | m9d0 | 290.0 | (257.8) | 179.0 | (75.1) | 518.0 | (16.7) |

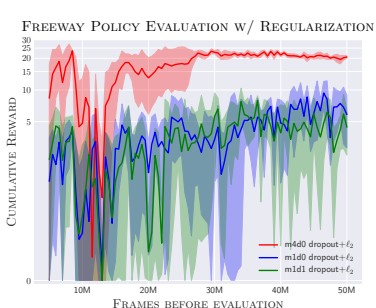

Table 2: Policy evaluation using regularization. Each game was initially trained in the default mode for 50M frames with dropout and $\ell_2$ regularization then evaluated on each listed flavour. Reported numbers are the average over 5 runs. Standard deviation is reported between parentheses.

Figure 3: Performance of an agent that was evaluated every $500,000$ frames after being trained in the default flavour of FREEWAY with dropout and $\ell_2$ regularization. Results are averaged over five seeds. The y-axis is log scaled.

In most flavours, evaluating the policy trained with regularization does not negatively impact performance when compared to the performance of the policy trained without regularization. In some flavours we even see an increase in performance. Interestingly, when using regularization the agent in FREEWAY improves for all flavours and even learns a policy capable of outperforming the baseline learned from scratch in two of the three flavours. Moreover, in FREEWAY we now observe increasing performance during evaluation throughout most of the learning procedure as depicted in Figure 3. These results seem to confirm the notion of overfitting observed in Figure 2.

Despite slight improvements from these techniques, regularization by itself does not seem sufficient to enable policies to generalize across flavours. As shown in the next section, perhaps the real benefit of regularization in deep RL comes from the ability to learn more general features. These features may lead to a more adaptable representation which can be reused and subsequently fine-tuned on other flavours, which is often the case in supervised learning.

# 6 VALUE FUNCTION FINE-TUNING

We hypothesize that the benefit of regularizing deep RL algorithms may not come from improvements during evaluation, but instead in having a good parameter initialization that can be adapted to new tasks that are similar. We evaluate this hypothesis using two common practices in machine learning. First, we the use the weights trained with regularization as the initialization for the entire network. We subsequently fine-tune all weights in the network. This is similar to what is performed in computer vision with supervised classification methods (e.g., Razavian et al., 2014). Secondly, we evaluate reusing and fine-tuning only early layers of the network. This has been shown to improve generalization in some settings (e.g., Yosinski et al., 2014), and is sometimes used in natural language processing (e.g., Mou et al., 2016; Howard & Ruder, 2018).

When fine-tuning the entire network, we take the weights of the network trained in the default flavour for 50M frames and use them to initialize the network commencing training in the new flavour for 50M frames. We perform this set of experiments twice. Once for the weights trained without regularization, and again for the weights trained with regularization, as described in the previous section. Each run is averaged over five seeds. For comparison we provide a baseline trained from scratch for 50M and 100M frames in each flavour. Directly comparing the performance obtained after fine-tuning to the performance after 50M frames (SCRATCH) shows the benefit of re-using a representation learned in a different task instead of randomly initializing the network. Comparing the performance obtained after fine-tuning to the performance of 100M frames (SCRATCH) lets us take into consideration the whole learning process. The results are presented in Table 3.

Fine-tuning from an unregularized representation yields conflicting conclusions. Although in FREEWAY we obtained positive fine-tuning results, we note that rewards are so sparse in m1d0 and m1d1 that this initialization is likely to be simply acting as a form of optimistic initialization, biasing the agent to go up. The agent observes rewards more often, therefore, it learns quicker about the new flavour. However, the agent is still unable to reach the maximum score in these flavours.

| GAME VARIANT | | FINE-TUNING | | | | REGULARIZED FINE-TUNING | | | | SCRATCH | | | |
|---|---|---|---|---|---|---|---|---|---|---|---|---|---|
| | | 10M | | 50M | | 10M | | 50M | | 50M | | 100M | |
| FREEWAY | m1d0 | 2.9 | (3.7) | 22.5 | (7.5) | 20.2 | (1.9) | 25.4 | (0.2) | 4.8 | (9.3) | 7.5 | (11.5) |
| | m1d1 | 0.1 | (0.2) | 17.4 | (11.4) | 18.5 | (2.8) | 25.4 | (0.4) | 0.0 | (0.0) | 2.5 | (7.3) |
| | m4d0 | 20.8 | (1.1) | 31.4 | (0.5) | 22.6 | (0.7) | 32.2 | (0.5) | 29.9 | (0.7) | 32.8 | (0.2) |
| HERO | m1d0 | 220.7 | (98.2) | 496.7 | (362.8) | 322.5 | (39.3) | 4104.6 | (2192.8) | 1425.2 | (1755.1) | 5026.8 | (2174.6) |
| | m2d0 | 74.4 | (31.7) | 92.5 | (26.2) | 84.8 | (56.1) | 211.0 | (100.6) | 326.1 | (130.4) | 323.5 | (76.4) |
| BREAKOUT | m12d0 | 11.5 | (10.7) | 69.1 | (14.9) | 48.2 | (4.1) | 96.1 | (11.2) | 67.6 | (32.4) | 55.2 | (37.2) |
| SPACE INVADERS | m1d0 | 617.8 | (55.9) | 926.1 | (56.6) | 701.8 | (28.5) | 1033.5 | (89.7) | 753.6 | (31.6) | 979.7 | (39.8) |
| | m1d1 | 482.6 | (63.4) | 799.4 | (52.5) | 656.7 | (25.5) | 920.0 | (83.5) | 698.5 | (31.3) | 906.9 | (56.5) |
| | m9d0 | 354.8 | (59.4) | 574.1 | (37.0) | 519.0 | (31.1) | 583.0 | (17.5) | 518.0 | (16.7) | 567.7 | (40.1) |

Table 3: Experiments fine-tuning the entire network with and without regularization (dropout + $\ell_2$). An agent is trained with dropout + $\ell_2$ regularization in the default flavour of each game for 50M frames, then DQN's parameters $\theta$ were used to initialize the fine-tuning procedure on each new flavour for 50M frames. The baseline agent is trained from scratch up to 100M frames. Standard deviation reported between parenthesis.

The results of fine-tuning the regularized representation are more exciting. In FREEWAY we observe the highest scores on m1d0 and m1d1 throughout the whole paper. In HERO we vastly outperform fine-tuning from an unregularized representation. In SPACE INVADERS we obtain higher scores across the board on average when comparing to the same amount of experience. These results suggest that reusing a regularized representation in deep RL might allow us to learn more general features which can be more successfully fine-tuned.

Moreover, initializing the network with a regularized representation has a big impact on the agent's performance when compared to initializing the network randomly. These results are impressive when we consider the potential regularization has in reducing the sample complexity of deep RL algorithms. Such an observation also holds when we take the total number of frames seen between two flavours into consideration. When directly comparing one row of REGULARIZED FINE-TUNING to SCRATCH we are comparing two algorithms that observed 100M frames. However, to generate two rows of SCRATCH we used 200M frames while two rows of REGULARIZED FINE-TUNING used 150M frames (50M from scratch + 50M in each row). The distinction becomes bigger and bigger as more tasks are taken into consideration.

We further investigate which layers may encode general features able to be fine-tuned. Inspiration was taken from other studies that have shown that neural networks can re-learn co-adaptations when their final layers are randomly initialized, sometimes improving generalization (Yosinski et al., 2014). We conjectured DQN may benefit from re-learning the co-adaptations between early layers comprising general features and the randomly initialized layers which ultimately assign state-action values. We hypothesized that it might be beneficial to re-learn the final layers from scratch since state-action values are ultimately conditioned on the flavour at hand. Therefore, we also evaluated whether fine-tuning only the convolutional layers, or the convolutional layers and the first fully connected layer was more effective than fine-tuning the whole network. Suprisingly, this does not seem to be the case. The performance obtained when the whole network is fine-tuned (Table 3) is consistently better than when it is not (Table 4). We speculate that this might not be the case on more dissimilar tasks.

## 7  DISCUSSION AND CONCLUSION

Many studies have tried to explain generalization of deep neural networks in supervised learning settings (e.g., Zhang et al., 2018; Dinh et al., 2017). Analyzing generalization and overfitting in deep RL has its own issues on top of the challenges posed in the supervised learning case. Actually, generalization in RL can be seen in different ways. We can talk about generalization in RL in terms of conditioned sub-goals within an environment (e.g., Andrychowicz et al., 2017; Sutton, 1995), learning multiple tasks at once (e.g., Teh et al., 2017; Parisotto et al., 2016), or sequential task learning as in a continual learning setting (e.g., Schwarz et al., 2018; Kirkpatrick et al., 2016). In this paper we evaluated generalization in terms of small variations of high-dimensional control tasks. This provides a candid evaluation method to study how well features and policies learned by

| GAME VARIANT | | REGULARIZED FINE-TUNING 3CONV | | | | REGULARIZED FINE-TUNING 3CONV+1FC | | | | REGULARIZED FINE-TUNING | |
|---|---|---|---|---|---|---|---|---|---|---|---|
| | | 10M | | 50M | | 10M | | 50M | | 50M | |
| FREEWAY | m1d0 | 0.0 | (0.0) | 0.7 | (1.4) | 0.1 | (0.1) | 4.9 | (9.9) | 25.4 | (0.2) |
| | m1d1 | 0.0 | (0.0) | 0.0 | (0.0) | 0.1 | (0.1) | 10.0 | (12.3) | 25.4 | (0.4) |
| | m4d0 | 7.3 | (3.5) | 30.4 | (0.6) | 4.9 | (4.8) | 30.7 | (1.7) | 32.2 | (0.5) |
| HERO | m1d0 | 405.1 | (82.0) | 1949.1 | (2076.4) | 350.3 | (52.1) | 3085.3 | (2055.6) | 4104.6 | (2192.8) |
| | m2d0 | 232.1 | (30.1) | 455.2 | (170.4) | 150.4 | (38.5) | 307.6 | (64.8) | 211.0 | (100.6) |
| BREAKOUT | m12d0 | 4.3 | (1.7) | 63.7 | (26.6) | 5.4 | (0.8) | 89.1 | (16.7) | 96.1 | (11.2) |
| SPACE INVADERS | m1d0 | 669.3 | (29.1) | 998.1 | (78.8) | 681.3 | (17.2) | 989.6 | (39.4) | 1033.5 | (89.7) |
| | m1d1 | 609.8 | (16.6) | 836.3 | (55.9) | 638.7 | (19.1) | 883.4 | (38.1) | 920.0 | (83.5) |
| | m9d0 | 436.1 | (18.9) | 581.0 | (12.2) | 439.9 | (40.3) | 586.7 | (39.7) | 583.0 | (17.5) |

Table 4: Experiments fine-tuning early layers of the network trained with regularization. An agent is trained with dropout + $\ell_2$ regularization in the default flavour of each game for 50M frames, then DQN's parameters $\theta$ were used to initialize the corresponding layers to be further fine-tuned on each new flavour. Remaining layers were randomly initialized. Compared against fine-tuning the entire network from Table 3. Standard deviation reported between parenthesis.

deep neural networks in RL problems can generalize. The approach of studying generalization with respect to the representation learning problem intersects nicely with the aforementioned problems in RL where generalization is key.

The empirical evaluation presented in this paper has shown that traditional DQN seems to generalize poorly even between very similar high-dimensional control tasks. Given this lack of generality we investigated how dropout and $\ell_2$ regularization can be used to improve generalization in deep RL. Other forms of regularization in RL that have been explored in the past are sticky-actions, random initial states, entropy regularization (Zhang et al., 2018), and procedural generation of environments (Justesen et al., 2018). More related to our work, regularization in the form of weight constraints has been applied in the continual learning setting in order to reduce the catastrophic forgetting exhibited by fine-tuning on many sequential tasks (Kirkpatrick et al., 2016; Schwarz et al., 2018). Similar weight constraint methods have been explored in multitask learning (Teh et al., 2017).

Evaluation practices in RL often focuses on training and evaluating agents on exactly the same task. Consequently, regularization has traditionally been underutilized in deep RL. With a renewed emphasis on generalization in RL, regularization applied to the representation learning problem can be a feasible method to improving generalization on closely related tasks. Our results suggest that dropout and $\ell_2$ regularization seem to be able to learn more general purpose features which can be adapted to similar problems. Although other communities relying on deep neural networks have shown similar successes, this is of particular importance for the deep RL community which struggles with sample efficiency (Henderson et al., 2018). This work is also related to recent meta-learning procedures like MAML (Finn et al., 2017) which aim to find a parameter initialization that can be quickly adapted to new tasks. In fact, some of the results here can also be seen under the light of curriculum learning. The regularization techniques we've evaluated here seem to be effective in leveraging situations where an easier task is presented first, sometimes leading to unseen performance levels (e.g., FREEWAY).

Finally, we believe it would be extremely beneficial for the field if we were able to develop algorithms that can generalize across tasks. Ultimately we want agents that can keep learning as they interact with the world in a continual learning fashion. The ability to generalize is essential. Throughout this paper we often avoided the expression *transfer learning* because we believe that succeeding in slightly different environments should be actually seen as a problem of generalization. Our results suggested that regularizing and fine-tuning representations in deep RL might be a viable approach towards improving sample efficiency and generalization on multiple tasks. It is particularly interesting that fine-tuning a regularized network was the most successful approach because this might also be applicable in the continual learning settings where the environment changes without the agent being told so, and re-initializing layers of a network is obviously not an option. In this setting, the work from Kirkpatrick et al. (2016), and Schwarz et al. (2018) might be a great starting point as they provide a more thorough discussion of generalization in continual learning.

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

## A  GAME MODES

FREEWAY

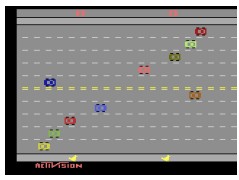
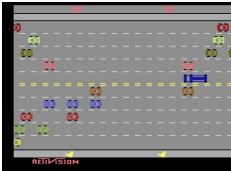
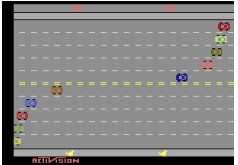

(a) FREEWAY m0d0       (b) FREEWAY m1d0       (c) FREEWAY m4d0

In FREEWAY a chicken must cross a road containing multiple lanes of moving traffic within a pre-specified time limit. In all modes of FREEWAY, the agent gets rewarded for reaching the top of the screen and is subsequently teleported to the bottom of the screen. If the chicken collides with a vehicle in difficulty 0 it gets bumped down one lane of traffic, alternatively, in difficulty 1 the chicken gets teleported to its starting position on the bottom of the screen. Mode 1 changes some vehicle sprites to include buses, adds more vehicles to some lanes, and increases the velocity of all vehicles. Mode 4 is almost identical to Mode 1; the only difference being vehicles can oscillate between two speeds.

HERO

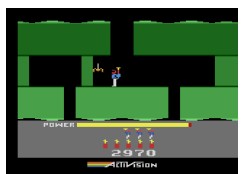
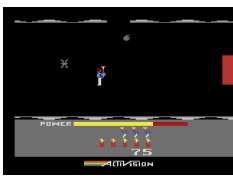
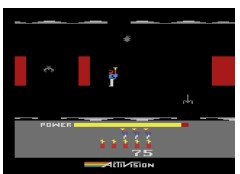

(a) HERO m0d0       (b) HERO m1d0       (c) HERO m2d0

In HERO you control a character who must navigate a maze in order to save a trapped miner within a cave system. The agent scores points for any forward progression such as clearing an obstacle or killing an enemy. Once the miner is rescued, the level is terminated and you continue to the next level with a different maze. Some levels have partially observable rooms, more enemies, and more difficult obstacles to traverse. Past the default mode, each subsequent mode starts off at increasingly harder levels denoted by a level number increasing by multiples of 5. The default mode starts you off at level 1, mode 1 starts at level 5, and so on.

BREAKOUT

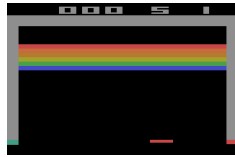
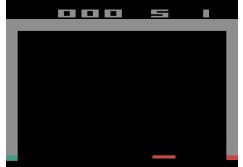

(a) BREAKOUT m0d0

(b) BREAKOUT m12d0

In BREAKOUT you control a paddle which can move horizontally along the bottom of the screen. At the beginning of the game, or on loss of life a ball is set into motion and can bounce off the paddle and collide with bricks at the top of the screen. The objective of the game is to break all the bricks without having the ball fall below your paddles horizontal plane. Subsequently, mode 12 of breakout hides the bricks from the player until the ball collides with the bricks in which case the bricks flash for a brief moment before disappearing again.

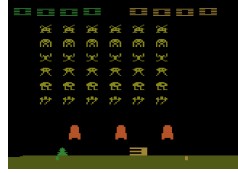 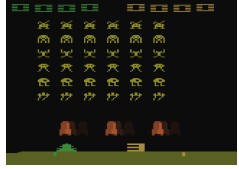 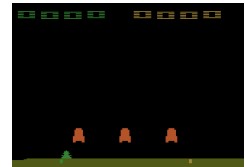

(a) SPACE INVADERS m0d0     (b) SPACE INVADERS m1d1     (c) SPACE INVADERWS m9d0

## SPACE INVADERS

When playing SPACE INVADERS you control a spaceship which can move horizontally along the bottom of the screen. There is a grid of aliens which are above you and the objective of the game is to shoot-out all aliens. You are afforded some protection from the alien bullets with three barriers just above the spaceship. Difficulty 1 of space invaders widens your spaceships sprite making it harder to doge enemy bullets. Mode 1 of SPACE INVADERS causes the shields above you to oscillate horizontally. Mode 9 of SPACE INVADERS is similar to Mode 12 of BREAKOUT where the aliens are partially observable until struck with the players bullet.

## B  BASELINE RESULTS

In all experiments performed in this paper we utilize the neural network architecture used by Mnih et al. (2015). That is, a convolutional neural network with three convolutional layers and two fully connected layers. A visualization of this network can be found in Figure 8. Hyperparametes are generally kept consistent with Machado et al. (2018). Below we provide a table of the key hyperparameters used in the baseline experiments.

NEURAL NETWORK ARCHITECTURE

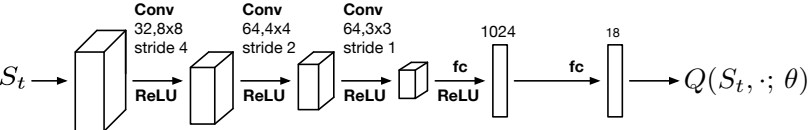

Figure 8: Neural network architecture used by DQN to predict state-action values.

HYPERPARAMETERS

| | | | |
|---|---|---|---|
| Learning rate $\alpha$ | 0.00025 | Replay buffer size | $1,000,000$ |
| Minibatch size | 32 | Target update frequency | 4 |
| Dropout rate convolutions | 0.05 | $\epsilon$ decay horizon | 1M frames |
| Dropout rate fully connected | 0.1 | $\epsilon$ initial | 1.0 |
| Regularization term $\lambda$ | 0.0001 | $\epsilon$ final | 0.01 |
| | | Discount factor $\gamma$ | 0.99 |

EVALUATION

Each baseline run is trained for up to 100M frames in each game flavour. We decay epsilon linearly over the $\epsilon$-decay period to allow for an exploratory period at the beginning of training. We use sticky-actions with a probability of $p = 0.25$ of executing $A_{t-1}$ instead of action $A_t$ (Machado et al., 2018). We allow the agent access to all 18 primitive actions in the ALE, we do not utilize the reduced action set nor the lives signal.

Furthermore, as a crude measure for environment complexity, we measure the best greedy action an agent could take in a game flavour. Simply put, we iterate through every action in $\mathcal{A}$, executing this action $\epsilon$-greeidly, with $\epsilon = 0.01$, at every time step for 100 episodes. These results were then averaged over 5 runs with the standard deviations between runs reported in parenthesis.

| GAME VARIANT | | 10M | | 50M | | 100M | | BEST ACTION | |
|---|---|---|---|---|---|---|---|---|---|
| FREEWAY | m0d0 | 3.0 | (1.0) | 31.4 | (0.2) | 32.1 | (0.1) | 23.0 | (1.4) |
| | m1d0 | 0.0 | (0.1) | 4.8 | (9.3) | 7.5 | (11.5) | 5.0 | (1.5) |
| | m1d1 | 0.0 | (0.0) | 0.0 | (0.0) | 2.5 | (7.3) | 4.2 | (1.3) |
| | m4d0 | 4.4 | (1.4) | 29.9 | (0.7) | 32.8 | (0.2) | 7.5 | (2.8) |
| HERO | m0d0 | 3187.8 | (78.3) | 9034.4 | (1610.9) | 13961.0 | (181.9) | 150.0 | (0.0) |
| | m1d0 | 326.9 | (40.3) | 1425.2 | (1755.1) | 5026.8 | (2174.6) | 75.8 | (7.5) |
| | m2d0 | 116.3 | (11.0) | 326.1 | (130.4) | 323.5 | (76.4) | 12.0 | (27.5) |
| BREAKOUT | m0d0 | 17.5 | (2.0) | 72.5 | (7.7) | 73.4 | (13.5) | 2.3 | (1.3) |
| | m12d0 | 17.7 | (1.3) | 67.6 | (32.4) | 55.2 | (37.2) | 1.8 | (1.1) |
| SPACE INVADERS | m0d0 | 250.3 | (16.2) | 698.8 | (32.2) | 927.1 | (85.3) | 243.6 | (95.9) |
| | m1d0 | 203.6 | (24.3) | 753.6 | (31.6) | 979.7 | (39.8) | 192.6 | (65.7) |
| | m1d1 | 193.6 | (11.0) | 698.5 | (31.3) | 906.9 | (56.5) | 180.9 | (101.9) |
| | m9d0 | 173.0 | (17.8) | 518.0 | (16.7) | 567.7 | (40.1) | 174.6 | (65.9) |

Table 5: Baselines using vanilla DQN for all tested game variants.

| GAME VARIANT | | 10M | | 50M | | 100M | | BEST ACTION | |
|---|---|---|---|---|---|---|---|---|---|
| FREEWAY | m0d0 | 4.6 | (5.0) | 25.9 | (0.6) | 29.0 | (0.8) | 23.0 | (1.4) |
| HERO | m0d0 | 2466.5 | (630.8) | 6505.9 | (1843.0) | 12446.9 | (397.4) | 150.0 | (0.0) |
| BREAKOUT | m0d0 | 6.1 | (2.7) | 34.1 | (1.8) | 66.4 | (3.6) | 2.3 | (1.3) |
| SPACE INVADERS | m0d0 | 214.6 | (13.8) | 623.1 | (16.3) | 617.4 | (29.6) | 243.6 | (95.9) |

Table 6: Baselines using dropout + $\ell_2$ regularization for each default flavour.

| | | BASELINE | | | | | | BASELINE W/ REGULARIZATION | | | | | |
|---|---|---|---|---|---|---|---|---|---|---|---|---|---|
| GAME VARIANT | | 10M | | 50M | | 100M | | 10M | | 50M | | 100M | |
| FREEWAY | m0d0 | 3.0 | (1.0) | 31.4 | (0.2) | 32.1 | (0.1) | 4.6 | (5.0) | 25.9 | (0.6) | 29.0 | (0.8) |
| HERO | m0d0 | 3187.8 | (78.3) | 9034.4 | (1610.9) | 13961.0 | (181.9) | 2466.5 | (630.8) | 6505.9 | (1843.0) | 12446.9 | (397.4) |
| BREAKOUT | m0d0 | 17.5 | (2.0) | 72.5 | (7.7) | 73.4 | (13.5) | 6.1 | (2.7) | 34.1 | (1.8) | 66.4 | (3.6) |
| SPACE INVADERS | m0d0 | 250.3 | (16.2) | 698.8 | (32.2) | 927.1 | (85.3) | 214.6 | (13.8) | 623.1 | (16.3) | 617.4 | (29.6) |

Table 7: Comparison of baseline results with and without regularization in the default flavour. The baseline agent with regularization was trained with dropout and $\ell_2$ regularization.

# C  POLICY EVALUATION LEARNING CURVES

We provide learning curves for evaluating a policy learned in the default flavour (m0d0) to each subsequent flavour of that game. Each subplot are the results of evaluating the policy from a representation trained with and without regularization.

## EVALUATION

Checkpoint of the network weights $\theta$ were taken during training every $500,000$ frames, up to 50M frames in total. Each checkpoint was then evaluated in the target mode for $100$ episodes averaged over five runs. Hyperparameters are kept consistent with the baseline experiments in Appendix B.

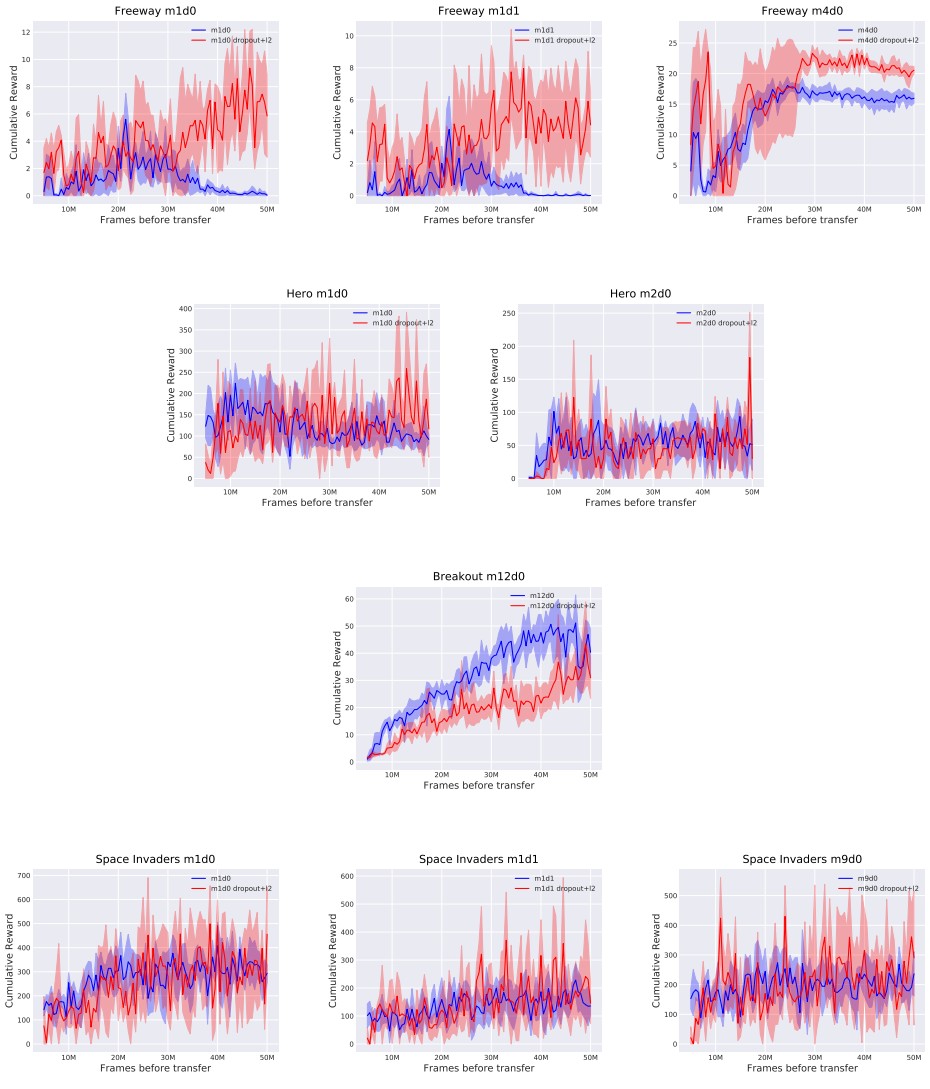

Figure 9: Performance curves for policy evaluation results. The x-axis is the number of frames before we evaluated the $\epsilon$-greedy policy from the default flavour on the target flavour. The y-axis is the cumulative reward the agent incurred.

