# OpenReview forum: "Generalization and Regularization in DQN"
_ICLR.cc/2019/Conference_

### Official Review · AnonReviewer3 · 2018-11-01
**Empirical Paper on Evaluating Generalization Properties with Regularized / Non-Regularized DQN**

**Rating:** 5
**Confidence:** 5

**Review:**

This is an empirical study on the ability for DQNs trained with/without regularization to perform well on variants of the same environment (e.g. increasing difficulty of a game). The paper is well written, the experimental methodology is clear & sound, and the significance is around improved sample efficiency via warm starting from a regularized DQN to fine tune. The error bounds for the regularized models results seem uncomfortably large in some cases. Overall it looks like a good methodological paper that can inform others on taking regularization more seriously when training DQNs. Evaluating on a modified ALE environment is great, but it would have been better to see this having similar impact in real life applications.

---

> ### Author Response · Authors · 2018-11-22
> **Reviewer 3 response**
>
> We would like to thank the reviewer for feedback on our work. We are pleased to see that the reviewer thinks our paper is “well written”, that its “experimental methodology is clear & sound”. We agree with the assessment that it is “good methodological paper that can inform others on taking regularization more seriously when training DQNs”, or even in a more general sense, that is a paper that raises awareness to this discussion about regularization/overfitting in deep RL. Because of all that, we were actually surprised by the score/rating we received. We do think this is an important paper, with an experimental methodology that is clear and sound, that raises awareness of important topics in the field.
> In the review we received, two concerns were raised: the “error bounds (...) seem uncomfortably large in some cases” and that it “would have been better to see this having similar impact in real-life applications”. We don’t think any of them are justified. Our error bounds were generated from 5 unique random seeds where most works utilize one or two seeds. Therefore, we actually used at least twice as many samples to generate those bounds. This is not a feature of a specific approach but in a sense of the whole field. Performance in Atari 2600 games have been extensively documented and it has high variance, for multiple reasons (stochasticity in the policy, dynamics of the environments, etc). We refer to Machado et al. (2018) for detailed results about the variance of different methods in dozens of Atari 2600 games. Finally, regarding the “real life applications”, we do agree works with such results are extremely important to the field. However, our paper should be evaluated on its merit, as well as what it does, which is to use a well-established framework for research in reinforcement learning.
>
> Reference:
> Marlos C. Machado, Marc G. Bellemare, Erik Talvitie, Joel Veness, Matthew J. Hausknecht, Michael Bowling: Revisiting the Arcade Learning Environment: Evaluation Protocols and Open Problems for General Agents. J. Artif. Intell. Res. 61: 523-562 (2018)

---

### Official Review · AnonReviewer1 · 2018-11-03
**Interesting study but need more experiments.**

**Rating:** 5
**Confidence:** 5

**Review:**

Summary:
This paper focuses on a "generalization" in deep Q-network (DQN). Specifically, they showed that when features (parameters of DQN) are trained in one environment (default flavour/mode) and then used as an initialization for the same model but for a slightly different environment ( i.e. still captures key concepts of the original environment ) can boost the performance of the model in the new flavour/environment. More importantly, the performance boost is significant when DQN's parameters which are used to initialize the model for the new environment were trained using dropout and L2 regularization in the default flavour/mode.  For the experiments, 4 games: FREEWAY, HERO, BREAKOUT, and SPACE INVADERS which have 13 flavours (combinations of a mode and a difficulty) are used.

Strengths:
+ This paper is interesting in the sense that it empirically shows that using regularization in training deep RL can be helpful when the goal is the generalization from one flavour of an environment to another one but very similar to the original.
+ The experiments show that using REGULARIZED FINE-TUNING and FINE-TUNING for a new flavour /mode can help with sample efficiency compared with the models which are trained from scratch [10M frames vs 50M frames and 50M frames vs 100M frames ](Table 3).
+ The paper is well-written and it can be easily followed.

Weaknesses:
- In my view, there should be experiments in which the proposed method is compared with other approaches that improve generalization in deep RL like Zhang et al., 2018 and Justesen et al., 2018.
- According to the paper (at least my understanding) DQN's hyper-parameters are tuned based on default mode/flavour environment. It is possible that results for 'SCRATCH' results (Table 3) can be improved if DQN's hyper-parameters are tuned on the current flavour not default one.
-The proposed method is only applicable when the default environment and the new one are very similar. If the environments are that similar why even bother to train first on default and then generalize to the new one? Wouldn't be less expensive to just focus on the target environment and find the best model on that?

Questions:
- It is mentioned in the paper, that evaluation protocol suggested by Machado et al. (2018) is being followed in this paper. Have you followed the protocol introduced in section 4.2 of Machado et al. (2018)? If yes, the protocol in Machado et al. (2018) is for training time but numbers in Table 1 in this paper are for evaluation time. Can you elaborate?
- Why only those 6 games were selected for the experiments?

In summary, I found this paper is interesting but my concern is about the experiments.

---

> ### Author Response · Authors · 2018-11-22
> **Reviewer 1 response**
>
> We would like to thank the reviewer for their time and useful feedback on our work. Three main weaknesses were raised: (i) the fact that we didn’t compare against other baselines, (ii) the fact we didn’t tune the hyperparameters for the new flavours, and (iii) the fact that the proposed method is only applicable between domains that are very similar. We don’t agree with this assessment and we don’t think any of these three issues are real weaknesses in our work.
>
> Regarding comparing to other baselines, we would first like to point out that our work is the first to perform such a comprehensive study using the different flavours of the Atari games supported by the ALE. Regardless of that, despite the fact that both mentioned papers are, to the best of our knowledge, preprint that were not peer-reviewed, we believe they are complementary to our work. When considering the work from Justesen et al. (2018), it is important to mention that there are no procedural generators for Atari 2600 games. Combining methods like Justesen et al.’s (2018) and Teh et al.’s (2017) to multitask learning is definitely an interesting avenue for future research though. We really like the work from Zhang et al. (2018)  but we also think it is not directly comparable to ours. They do perform a careful study on possible sources of overfitting in RL, and how effective are approaches to avoid it such as using stochastic policies, null-ops, and human restarts, or sticky actions. This is obviously very different from what we did and from the discussion we are raising. As before, it is complimentary.
>
> The comment about fine-tuning the hyperparameters to the other domains exactly highlights one of the concerns we raise in the paper. It is not practical. It poses a huge computational burden to the practitioner and we are advocating exactly against it. We believe our methods should be robust enough to be applied to slightly different problems, without requiring a complete grid search every time a new task shows up. Our results suggest that regularizing the network being trained is a promising step towards that direction.
>
> Finally, we agree that when tasks are very different it is unlikely that regularizing the network would work. However, it is important to notice that when tasks are very similar, training from scratch on each desired environment is very computationally expensive. Deep RL already has issues with sample complexity and large computational burdens which acts as a motivation for this work. The scenario where there are multiple similar tasks is not unrealistic, as it is often studied in multi-task and continual learning settings as well.
>
> To conclude, two questions were asked, (1) about the evaluation protocol and (2) about the number of games being used. We appreciate the confusion from the evaluation protocol and we will make sure to clarify it in the final version of the paper. We did follow the evaluation protocol outlined in Section 4.2 of Machado et al., (2018). To be specific, we report the performance during training as the average cumulative reward over the previous 100 episodes. In Table 1, the EVALUATION column represents the average cumulative reward over 100 episodes in the new flavor for the policy learned in the default flavor. Only in that case, because there was no training, we simply evaluated the policy; however, it is important to keep in mind the same principle is being used, as we are still looking at the *current* performance of the agent after seeing that many frames. Finally, about the number of games being used, we evaluate 13 different flavors for up to 100M frames. In doing we ran 5 seeds to reduce the error bounds. Often only one or two seeds are used, which means that we used at least twice as much computation to generate a similar result in order to have a better reliability. We would have liked to use more games and flavours but we were bound by the computational resources we have available.

---

### Official Review · AnonReviewer2 · 2018-11-04
**Not unknown but nice systematic exploration**

**Rating:** 6
**Confidence:** 3

**Review:**

I totally disagree with the authors that any of their observations are surprising. Indeed the fact that an RL agent does not generalizes to small modifications of the task (either visual or in the dynamics) is well known. If the agent should generalize though is a different question. And I do not mean this in the sense that it is an undesirable property but rather if it is outside of what “learning one task” means. Particularly I feel this is a very pessimistic view of RL and potentially not even in-line with what happens in supervised learning.

I think one mantra of deep learning (and deep RL needs to obey by it) is that one should test in the same setting as training for things to truly work. For supervised, there is a distribution of data, and the test set are samples from the same distribution. However the testing scenario used here is slightly different. During training, if I do not see car accelerating, I think it makes no sense to expect to generalize to a new game that has this property as it is out-of-distribution. Of course it would be ideal if it could do that. And to clarify, while for us some of these extensions seem very similar and minimal changes, hence it should generalize to rather than transfer to, this is just the effect of imposing our own biases on the learning process. Deep Nets do not learn like we do, and in their universe they have never seen a car accelerating -- it makes sense that it might not to be able to generalize to it. Again, I’m not arguing that we don’t want this, but rather if we should expect it as part of what the system should normally generalize to.

To that end I think this paper enters in that unresolved dispute of what generalization should be versus what is transfer. At what point do we have truly a new task vs a sample from the same task. I don’t think there is an answer.

Going back to the observations in this work. I think the fact that the environment is not stochastic reinforces this overfitting (as in the extreme you end up with a policy that just repeats the optimal sequence of actions). I think in this particular case I can see how finetuning to a variation of the task fails. However true stochasticity in the environment (e.g. having a distribution of variations) like is done in Distral paper (where each episode is a different layout) can behave as a regularizer that will mitigate a bit the overfitting. That is to say that I believe the observed behaviour will be less pronounced in complex stochastic setting.

Nevertheless the paper seems to highlight an important observation (and back it up with empirical evidence), namely we should use more regularization like L2 or otherwise in practice. Which is mostly absent from publications. And I think this on its own is valuable.

---

> ### Author Response · Authors · 2018-11-22
> **Reviewer 2 response**
>
> We would like to thank the reviewer for their time and useful feedback on our work. We understand the perspective from which the results we presented are not surprising. We like the distinction made by the reviewer about the fact that it is expected that the agent generalizes to different domains, although current approaches don’t allow that. This is a discussion that is important and our goal is that our paper can contribute to raising awareness of the problem that our current algorithms lack the ability to generalize the slight variations of tasks. Ideally, our agents should be able to do so, although they are not right now. We are happy to de-emphasize the discussion about the results being “surprising” under a technical point of view.
>
> Finally, we would just like to point out that the environments we used are *not* deterministic. We used the new version of the ALE with sticky actions (see Machado et al, 2018) where,
> with a given probability (25%), we have a 1/4 chance of selecting the previous action instead of the action the agent wants to execute at the current timestep. Obviously, this is not a huge source of stochasticity, but Machado et al. (2018) have shown that it is enough to break algorithms that heavily rely on the determinism of the environment. Extending this work to other domains is definitely an interesting future research direction.
>
>
> Reference:
> Marlos C. Machado, Marc G. Bellemare, Erik Talvitie, Joel Veness, Matthew J. Hausknecht, Michael Bowling: Revisiting the Arcade Learning Environment: Evaluation Protocols and Open Problems for General Agents. J. Artif. Intell. Res. 61: 523-562 (2018)

---

### Meta-Review · Area_Chair1 · 2018-12-15

**Confidence:** 5
**Recommendation:** Reject

**Metareview:**

The authors have presented an empirical study of generalization and regularization in DQN. They evaluate generalization on different variants of Atari games and show that dropout and L2 regularization are beneficial. The paper does not contain any major revelations, nor does it propose new algorithms or approaches, but it is a well-written and clear demonstration, and it would be interesting to the deep RL community. However, the reviewers did not feel that the paper met the bar for publication at ICLR because the experiments were not more comprehensive, which would be expected for an empirical study. The AC will side with the reviewers but hopes that the authors will expand their study and resubmit to another venue in the future.